# Intolerance upon statin rechallenge: A systematic review and meta-analysis of randomized controlled trials

**Roni Kraut**[1]*, **Faith Wierenga**[2], **Elisa Molstad**[3], **Christina Korownyk**[1], **Danielle Perry**[1,4], **Liz Dennett**[5], **Scott Garrison**[1]

1 Department of Family Medicine, University of Alberta, Edmonton, Canada, 2 Faculty of Medicine and Dentistry, University of Alberta, Edmonton, Canada, 3 Faculty of Science, University of Alberta, Edmonton, Canada, 4 College of Family Physicians of Canada, Mississauga, Ontario, Canada, 5 Sperber Health Sciences Library, University of Alberta, Edmonton, Canada

* rkraut@ualberta.ca

## Abstract

### Background

Although statins are often discontinued when myalgia arises, a causal relationship may not always exist. How well-tolerated statins are when rechallenge is blinded and controlled is unclear.

### Methods and findings

We performed a systematic review and meta-analysis (PROSPERO CRD42023437648) to evaluate the success of statin rechallenge versus matched placebo in those who were previously statin intolerant. Our primary outcome was intolerance; our secondary outcome was the myalgia or global symptom score. Medline, Embase, CINAHL Plus, Scopus, and CENTRAL were searched from inception to May 1, 2023. Eligible trials were randomized controlled trials with parallel or crossover designs examining statin rechallenge in statin-intolerant adults. Two independent reviewers selected studies, extracted data, and assessed risk of bias (Cochrane Collaboration's risk-of-bias tool 1). Relative risk (RR) and mean difference (MD) were estimated using fixed effect Mantel-Haenszel statistics. Of 1,941 studies screened, 8 met our inclusion criteria (8 to 491 participants from Asia, Europe, North America, and Oceana). Compared to placebo, intolerance was more common in statin users [325/906 (36%) vs 233/911 (26%), RR 1.40, 95% CI, 1.23 to 1.60, $I^2$ = 0%, 7 trials, number needed to harm 10] and there was no statistically significant difference in myalgia or global symptom score on a 100-point scale [MD 1.08, 95% CI, -1.51 to 3.67, $I^2$ = 0%, 5 trials]. Limitations include only 1 trial asking participants about intolerable symptoms (vs inferring intolerance from discontinuation or trial withdrawal); the small number of trials; the possibility of attrition bias; and the potential for carryover effects in crossover/n-of-1 trial designs.

### Conclusions

Of those previously intolerant of statins who were rechallenged with a statin and compared to placebo recipients, medication intolerance was more common amongst statin recipients.

**Data Availability Statement:** All relevant data are within the paper and its Supporting Information files.

**Funding:** The authors received no specific funding for this work.

**Competing interests:** The authors declared that no competing interests exist.

However, there was no significant difference in mean myalgia or global symptom score between statin and placebo, and only one-third of those previously believed to be statin intolerant were unable to tolerate a statin on blinded rechallenge; one-quarter were intolerant of placebo.

## Introduction

The HMG-CoA reductase inhibitors class of medication, otherwise knowns as statins, are the first-line pharmaceutical option recommended to reduce the risk of cardiovascular disease in adults [1–3]. However, upwards of 10% of individuals experience symptoms while taking statins and are unable to continue on the medication [4].

Current evidence suggests that a large proportion of symptoms when on statins is secondary to the expectation of symptoms rather than true symptoms. This is commonly referred to as the nocebo effect; it is similar to the placebo effect, but rather than experiencing benefit from the placebo, the individual experiences perceived adverse effects. A randomized controlled trial found 90% of symptoms while on a statin also occurred while on placebo [5]. Furthermore, a recent systematic review found this occurs in 38% to 78% of individuals reporting statin-associated muscle symptoms [6].

This evidence implies that individuals who experience symptoms from statins may be able to successfully resume statin therapy, but to our knowledge there has not yet been a systematic review on this topic. The purpose of this systematic review was to evaluate whether statin-intolerant individuals can successfully resume statin use when re-introduction is blinded; the primary outcome was intolerance or discontinuation of a statin; and secondary outcome was the myalgia and if not available global symptom score (consolidation of symptoms potentially related to statins including myalgia).

## Methods

This systematic review is reported according to PRISMA (Preferred Reporting Items for Systematic Reviews and Meta-Analyses) [7]. See S1 Table for the completed PRISMA checklists.

### Data sources and searches

A health sciences librarian (LD) developed a search strategy and adapted it for Medline, Embase, CINAHL Plus, Scopus, and the Cochrane CENTRAL trial registry (S1 Text). The search was carried out on May 1, 2023, with no restrictions on date or language. We additionally searched for eligible trials using Google; trial registries (ClinicalTrials.gov and International Clinical Trials Registry Platform); and the reference lists of all included trials (FW). Covidence software was used to coordinate the review, and the protocol was published on the Prospero database (CRD42023437648) prior to data extraction.

### Study selection

Two authors (FW and EM) independently screened titles and abstracts and examined full texts to determine eligibility. Any conflicts were resolved by consensus or by a third reviewer (RK) if needed. Eligible studies were randomized controlled trials with a parallel or crossover/n-of-1 design that examined re-introduction of statin in adults previously considered to be intolerant. Valid comparators included matched placebo, usual care, no intervention, or a matched active

agent which was also provided to the intervention group. Trials were eligible if they included any of our outcomes, which include statin intolerance (primary outcome) and myalgia or global symptom score, transformable to a 100-point scale.

## Data extraction and quality assessment

Two authors (FW and EM) independently extracted data using Google spreadsheet. Differences were resolved by consensus or by involving a third person (RK). Authors were contacted to ask if our outcomes were available when not provided in the publication. Given that statin intolerance has no consensus definition [8], we preferentially used each trial's definition of intolerance, where available, but additionally accepted stopping of the study drug, or withdrawing from the study (due to myalgia or adverse events), when the definition of intolerance was not provided. When multiple potential measures were available, we determined statin intolerance according to the following predefined hierarchy:

1. The definition of intolerance provided in the trial

2. Composite of stopping the study drug due to myalgia OR withdrawing from the study due to myalgia (if these appear mutually exclusive)

3. Composite of stopping the study drug due to adverse effects OR withdrawing from the study due to adverse effects (if these appear mutually exclusive)

4. Stopping the study drug due to myalgia

5. Stopping the study drug due to adverse effects

6. Withdrawing from the study due to myalgia

7. Withdrawing from the study drug due to adverse effects

For myalgia, we accepted any continuous (visual analogue) or 11-point interval (0–10) myalgia score that was convertible to our 100-point scale. When myalgia was not specifically available, we also accepted a global symptom score. When data was available from multiple time points, we chose the furthest time point in the trial where attrition bias appeared negligible (defined as stopping and withdrawal being less than 10% in all groups). If stopping and withdrawal exceeded 10% at all time points in any group, that trial was excluded. We excluded data with high attrition bias given that those who were more symptomatic would be more likely to stop the study drug or withdraw from the trial, which would make both groups appear more similar than is reality. We predefined a minimally clinically important difference as 10 or greater on a 100-point symptoms scale [9].

Two data extractors (FW and EM) assessed all included trials for potential bias using the Cochrane Collaboration's risk-of-bias tool 1 [10]. Any differences were resolved by consensus or by a third person (RK). RevMan 5 software was used to generate the risk-of-bias figure [11].

## Data synthesis

A summary RR of intolerance (dichotomous) was estimated using Mantel-Haenszel methods and assuming fixed effects, which were chosen over random effects given the similarity in the intervention, control, and participant demographics. To ensure intolerance was not impacted by the potential nocebo effect, any eligible trials with unblinded participants or comparators of usual care and no intervention were not included in data synthesis. Heterogeneity was assessed using the $I^2$ statistic.

We predefined a subgroup analysis based on "lesser" and "greater" statin exposure to examine if intolerance was impacted by degree of statin exposure. We allocated trials into the lessor and greater exposure groups using median split of the length of each trials' individual study period (ie, the uninterrupted time on study drug). Thereby, trials with a period length less than the median were allocated to the lessor exposure group, and trials with a period length greater than the median were allocated to the greater exposure group. In the event of two trials having a period length equal to the median, or an odd number of included trials, we allocated such median straddling trials to greater or lesser exposure using a median split of the mean statin dose/potency (if such could be clearly determined) or, failing that, by the total time on study drug when all periods are combined.

We also produced a summary MD for myalgia or global symptom score, again using Mantel-Haenszel statistics with fixed effects. RevMan 5 software was used to complete the meta-analysis.

## Results

### Study selection

A total of 1,971 studies were screened for inclusion, with 53 undergoing full text review and 8 meeting our inclusion criteria (Fig 1). Table 1 provides key attributes of included trials. Our full data extraction spreadsheet is available in S2 Table. Overall, the 8 included trials were published between 2008–2021 and conducted in the United States (n = 2), United Kingdom (n = 2), Norway (n = 1), Canada (n = 1), and in multiple countries with a United States coordinating centre (n = 2). Only two trials reported funding from the pharmaceutical industry. All 8 trials were either exploring myalgia/intolerance upon rechallenge as the main trial objective (n = 6) or attempting to identify truly statin-intolerant subjects as an inclusion criterion for subsequent exploration of other therapies (n = 2).

Only 1 trial had a parallel design, with the others being n-of-1 (n = 3) or crossover (n = 4) The n-of-1 trials had 6–12 periods, ranging from 3 to 8 weeks long, none had a washout prior to study start, and one had a 3-week washout between periods. The crossover trials had 2 periods, ranging from 7 to 12 weeks long, the washout prior to study start ranged from none to 4 weeks, and the washout in between study periods ranged from none to 4 weeks. The parallel trial was 12 weeks long. The statins used were atorvastatin (n = 4), rosuvastatin (n = 1), fluvastatin (n = 1), simvastatin (n = 1), and multiple statins (n = 1); all but 1 study used a medium-intensity dose. All participants were blinded. A definition of statin intolerance was provided for only 2 trials.

Symptoms measured included myalgia (n = 7) or a global symptom score (n = 1); these were recorded using either the visual analogue score (VAS) (n = 3); brief pain inventory–short form (BPI–SF) (n = 2); VAS and BPI-SF (n = 1); a self-developed questionnaire (n = 1); or did not indicate the specific scale (n = 1). Participants recorded scores daily (n = 2), weekly (n = 3), every 2–4 weeks (n = 1), or the assessment time was not given (n = 2). In two trials scores were only recorded at the end of the period. In Herrett, scores were recorded daily the last week of each 8-week period, and in Kristiansen, scores were recorded weekly the last 3 weeks of each 7-week period. Four of 6 authors contacted were able to provide additional details.

The number of participants ranged from 8 to 491, and the median age was 65 years old, with an interquartile range of 61 to 66 years old. Participants were recruited from a variety of sites including primary care clinics, specialty clinics, hospitals, and a veteran affairs primary care centre. Approximately 50% of participants were male except Joy (13% male) and Kennedy (100% male); Kennedy was in a US Veteran Affairs Medical Centre. Among the 6 trials that reported ethnicity, all participants were ≥ 90% Caucasian except for Kennedy with 68% Caucasian, 18%

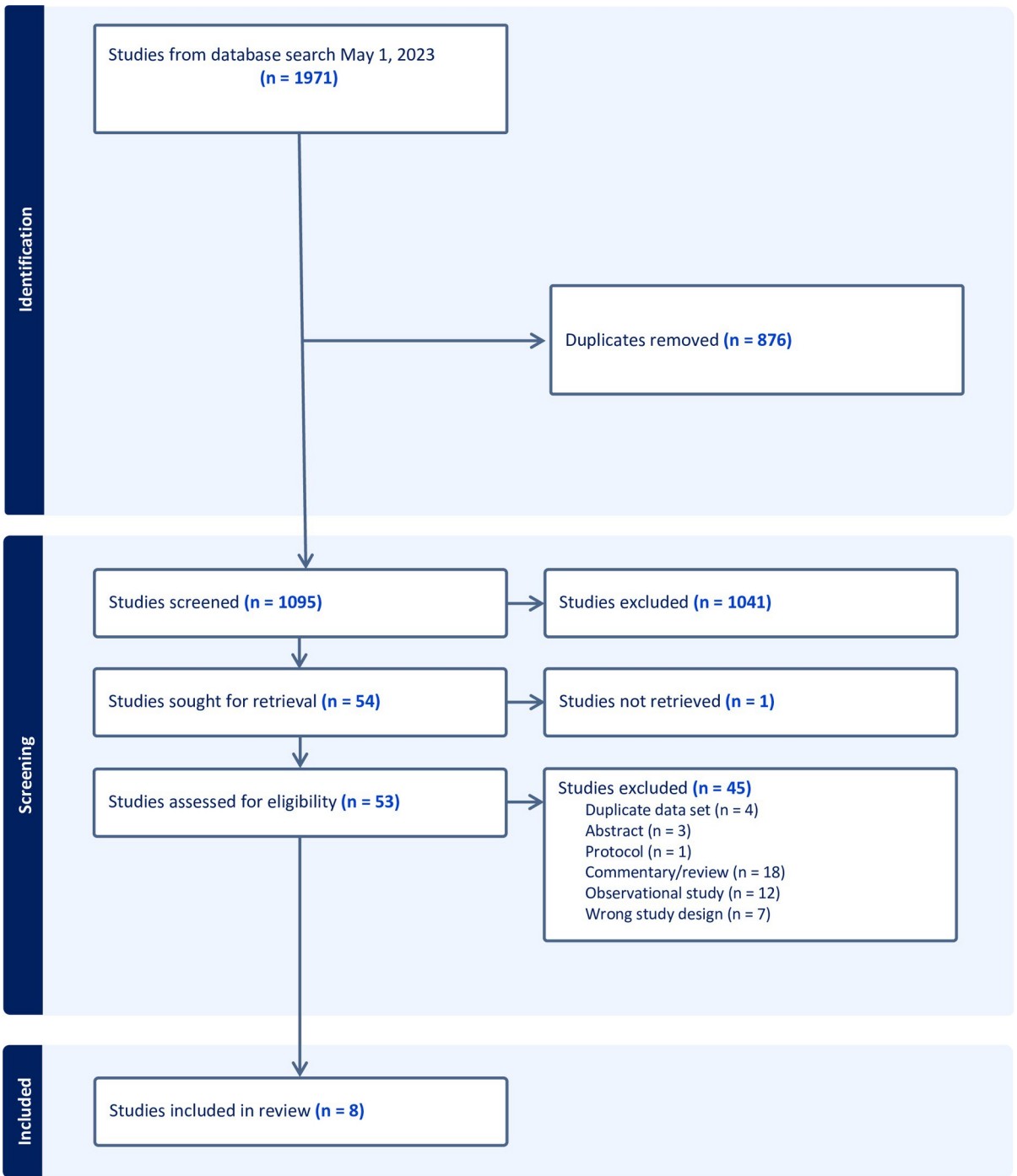

**Fig 1. Flow diagram of study selection process.**

African American, and 6% Hispanic. In 5 trials, >75% of participants discontinued statins prior to the trial due to myalgia or global symptoms attributed to statins; 2 trials stated some participants discontinued but did not provide a percentage; and 1 trial did not provide this data. In 3 trials, ≥75% of participants tried at least 2 statins before the trial; in 2 trials, 25% to 50% tried at least 2 statins; and the remainder of the trials did not provide this information.

**Table 1. Characteristics of included trials.**

| Study | Country | Design | Focus | Statin (intensity) | Trial periods & washout between periods | Intolerance outcome | Symptoms scale (symptom measured) | N | Recruitment site, tried ≥2 statins (%) |
|---|---|---|---|---|---|---|---|---|---|
| Herrett 2021 [12] | UK | N-of-1 | Rechallenge | Atorvastatin (Moderate) | 6, 8-week periods Washout: none | Withdrawals | VAS (myalgia) | 200 | 50 general practices, not given |
| Howard 2021 [5] | UK | N-of-1 | Rechallenge | Atorvastatin (Moderate) | 12, 4-week period Washout: none | Stops & withdrawals | VAS (global) | 60 | 17 referral centres and self-referral, 78% of participants |
| Joy 2014 [13] | Canada | N-of-1 | Rechallenge | Atorvastatin Rosuvastatin Pravastatin (Moderate) | 6, 3-week periods Washout: 3 weeks | Stops & withdrawals | VAS, BPI–SF (myalgia) | 8 | Endocrinology clinics & ads, ≥75% of participants |
| Kennedy 2011 [14] | USA | Crossover | Rechallenge | Rosuvastatin (Moderate) | 2, 8-week periods Washout: none | Stops & withdrawals | Not given (myalgia) | 17 | Veteran affairs primary care patients, 41% of participants |
| Kristiansen 2021 [15] | Norway | Crossover | Rechallenge | Atorvastatin (High) | 2, 7-week periods Washout: 1 week | VAS score[1] | VAS (myalgia) | 77 | Coronary heart disease discharges from 2 hospitals, 27% of participants |
| Nissen 2016 [16] | Multi-centre | Crossover | Evolocumab efficacy and tolerability | Atorvastatin (Moderate) | 2, 10-week periods Washout: 2 weeks | Self-reported symptoms | BPI–SF (myalgia) | 491 | Sites worldwide, 100% of participants |
| Stein 2008 [17] | Multi-centre | Parallel | Fluvastatin efficacy and tolerability | Fluvastatin (Moderate) | 1, 12-week periods Washout: NA | Stops & withdrawals | Self-developed questionnaire (myalgia) | 130[2] | Sites worldwide, not given |
| Taylor 2015 [18] | USA | Crossover | Coenzyme Q!0 impact on myalgia | Simvastatin (Moderate) | 2, 8-week periods Washout: 4 weeks | Not given | BPI–SF (myalgia) | 131 | Lipid clinics, ads & physician offices, not given |

NA–not applicable

[1] Defined as ≥25% higher individual mean VAS-score during the treatment period on atorvastatin vs. placebo, and ≥1 cm absolute difference.

[2] Total participants in this trial is 199 (3 arms), total participants in the two arms included in this systematic review is 130.

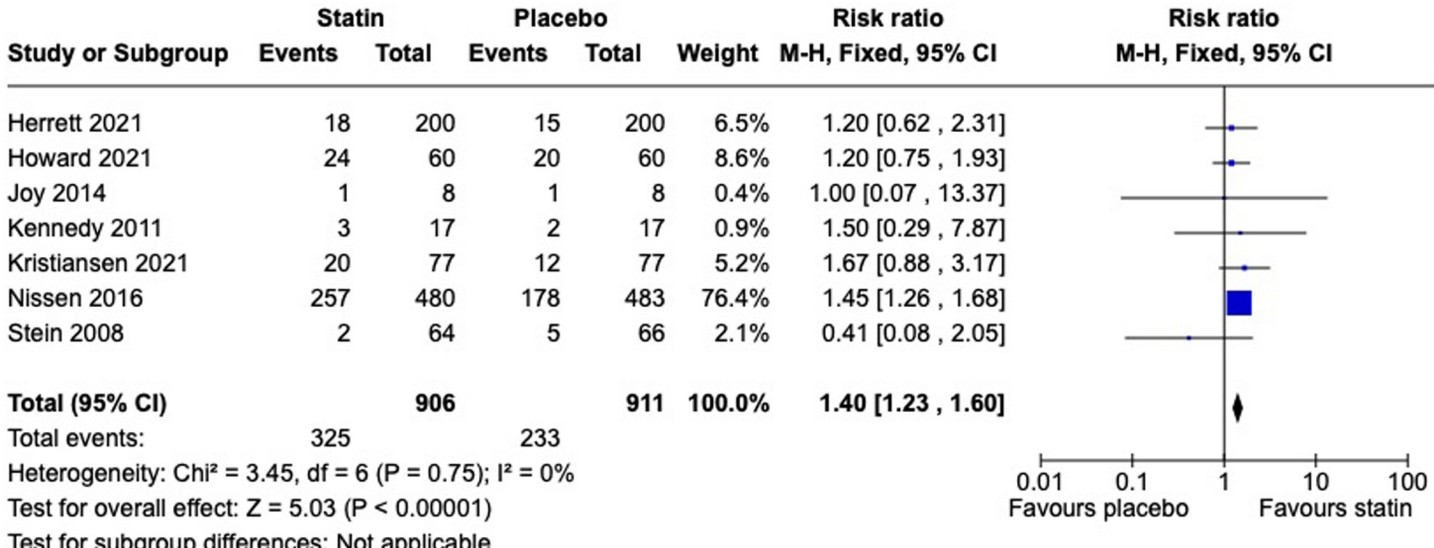

**Fig 2. Meta-analysis of intolerance on statin vs. placebo.**

## Intolerance

Seven trials provided data related to intolerance including patient-reported intolerable myalgia (n = 1); predefined change in VAS score (n = 1); stops and withdrawals (n = 4); and withdrawals (n = 1). In participants rechallenged with statin therapy, 36% experienced intolerance (325 events /906 total events), compared to 26% on placebo (233 events/911 total events) (RR 1.40, 95% CI, 1.23 to 1.60, $I^2$ = 0%, number needed to harm 10). This meta-analysis is shown in Fig 2. Risk of bias for the assessment of intolerance is shown in Fig 3.

These findings were similar between "lesser" and "greater exposure" subgroups (S1 Fig), which had a median split at a period of 8 weeks of statin rechallenge. The RR was also similar when we conducted a sensitivity analysis by excluding the two trials with intolerance based on participants symptoms, which accounted for slightly more than half of the participants (S2 Fig).

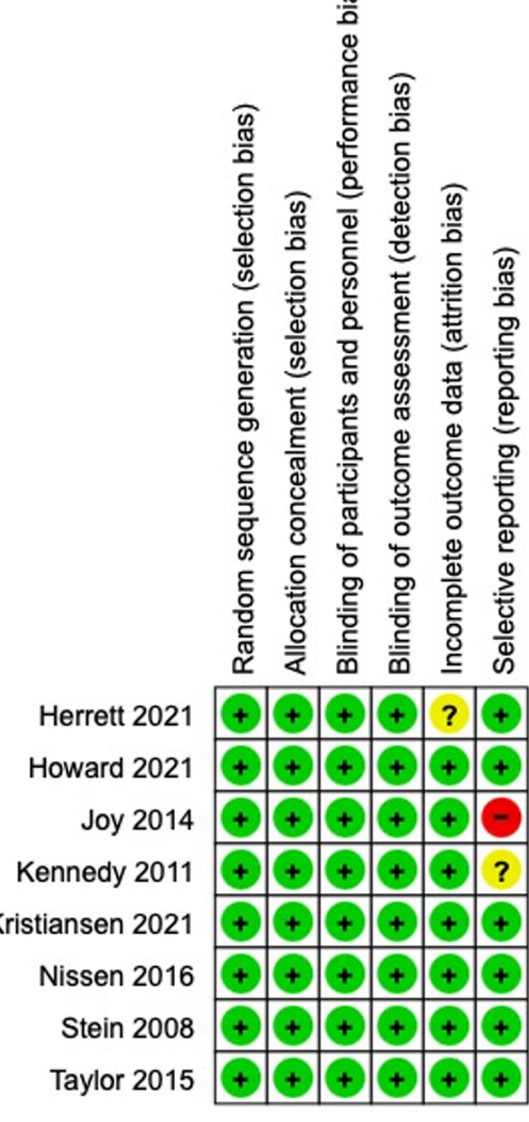

**Fig 3. Risk of bias assessment for intolerance outcome.**

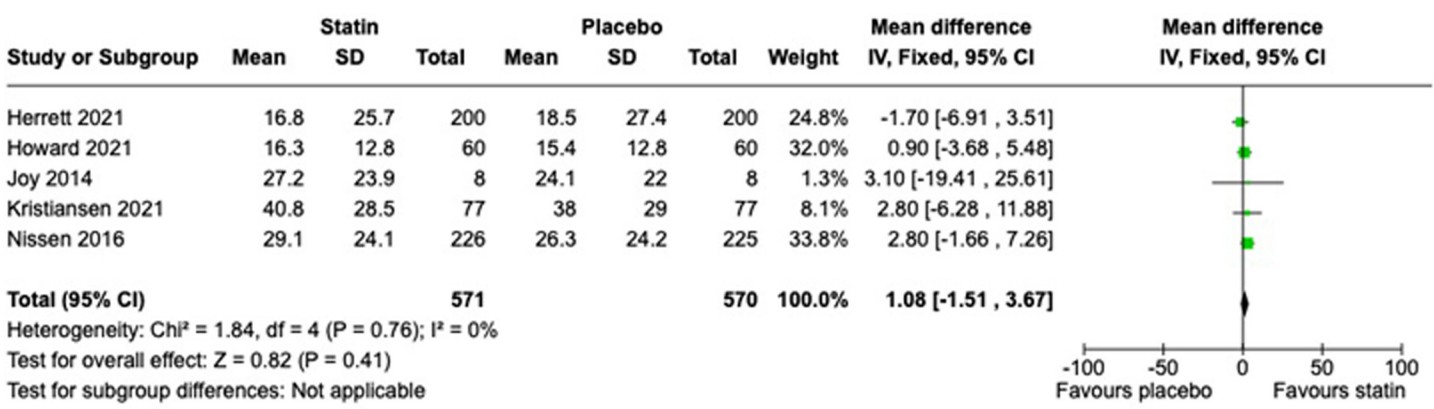

**Fig 4. Meta-analysis of mean myalgia or global symptom score on statin versus placebo.**

Two trials provided additional information relevant to intolerance either upon or after trial completion. This included Howard, where 30/60 (50%) remained on statins at 6 months, and Joy, where 5/8 (62.5%) remained on statins at 10 months.

## Myalgia or global symptom score

Five trials provided data for this analysis, four reporting myalgia and one reporting global symptoms. Overall, statin recipients were more symptomatic than placebo recipients, but this was not statistically significant: symptom scores were a mean 1.08 points higher out of 100 (95% CI, -1.51 to 3.67, $I^2$ = 0%). This meta-analysis is shown in Fig 4. Risk of bias for the assessment of myalgia or global symptoms score is shown in Fig 5.

## Discussion

We conducted a systematic review and meta-analysis of 8 randomized trials evaluating blinded rechallenge of statins in those previously believed to be intolerant. Only 36% of participants were intolerant of statins on rechallenge compared to 26% who were intolerant on placebo. The 2 trials that examined statin use 6–10 months after study completion suggest tolerance did not change substantially over time. In addition, no significant difference in mean myalgia or global symptom scores was found between statins and placebo. Further even if the upper bound on the 95% confidence interval (3.67 on a 100-point scale) is the true effect, this is still well below our predefined 10-point minimum clinically important difference.

These findings appear robust; risk of bias was low, our findings were consistent with sub-group and sensitivity analysis, and heterogeneity was low even with the variability in trial design. Generalizability is further improved with the wide variety of statins used; the variety of countries in which the trials took place (although most were Caucasian and in their 60s); the wide variety of preventive settings out of which participants were recruited; and the number of times participants attempted to take a statin prior to being enrolled in a trial.

Statins are first-line interventions for cardiovascular risk reduction and recommended in many clinical settings. Our findings suggest many individuals believed to be statin intolerant can successfully resume a statin when the expectation of adverse effects is lessened, as is the case with blinding and the possibility of placebo. Although an n-of-1 placebo-controlled trial may not be practical in the clinic, prescribers might achieve similar success by switching statins and using lower doses or alternate daily dosing [19, 20]. In addition, n-of-1 unblinded trials in clinic may be effective if prescribers provide reasonable expectation and counselling [21].

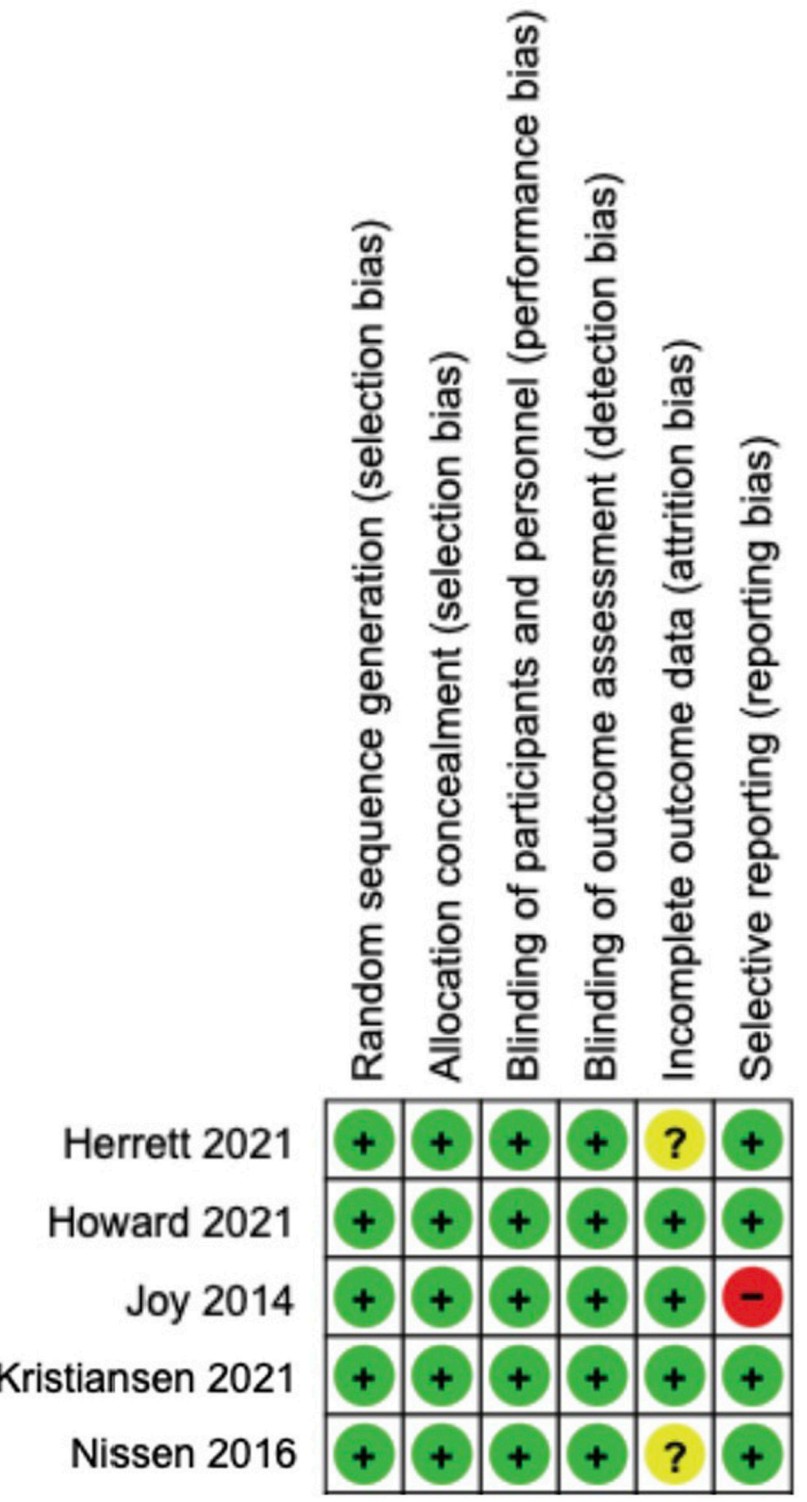

**Fig 5. Risk of bias assessment for myalgia or global symptom score outcome.**

Tudor et al. conducted a n-of-1 statin rechallenge trial that included a blinded and non-blinded arm, both receiving counselling from a primary care physician, and there was little difference in symptoms between arms [22].

Although the half-life of statin medication ranges between 5 and 30 hours [23], observational studies have found it can take several months for statin-related symptoms to develop and again several months for them to resolve [24–26]. Given the relatively shorter length of statin, placebo, and washout periods in all the included trials, there is potential for carryover effects that could make statin and placebo groups look more similar. Arguing against this occurring is the low overall rate of intolerance in the statin group and the success at continuing statins at 6 to 10 months in 2 trials with post-trial follow-up. We may have missed participants that were intolerant by using stopping and withdrawal from the study as alternative definitions of intolerance when intolerance was not provided. Mitigating against this is the high weighting of the trial asking about intolerable symptoms, the lack of heterogeneity between trials, and sensitivity analysis that found little difference between groups. There is a possibility of attrition bias, but doubtful this would cause a clinically meaningful difference to myalgia or global symptom score given that either trials adjusted the scores for attrition, had minimal attrition, or had similar attrition between statin and placebo periods. Lastly, there were only 8 trials included in this systematic review, however bias was low overall and over 900 unique individuals were randomized.

## Conclusions

Of those previously believed to be statin intolerant, only one-third will be intolerant of statins when introduced in a blinded, placebo-controlled rechallenge, and one-quarter of such individuals will be intolerant of placebo in the same setting. Clinicians should consider looking for ways to re-introduce statins (perhaps changing statins or lowering doses) before moving to second-line agents.

## Supporting information

**S1 Text. Search strategy.**
(PDF)

**S1 Table. PRISMA checklist.**
(PDF)

**S2 Table. Full data extraction spreadsheet.**
(XLSX)

**S1 Fig. Subgroup analysis—exposure.**
(PDF)

**S2 Fig. Subgroup analysis—statin intolerance definition.**
(PDF)

## Acknowledgments

We would like to thank Prof. MD, PhD John Munkhaugen and MD, PhD Oscar Kristiansen, Department of Medicine, Drammen Hospital, for providing additional data on the MUSE trial; Dr. Steven Nissen from Cleveland Clinic for providing additional data on the GAUSS-3 trial; Emily Herrett and Alexander Perkins from the London School of Hygiene and Topical Medicine for providing clarification on the statinWISE trial; and Dr. James Howard and Professor Darrel Francis from Imperial College London for providing clarification on the SAMSON trial.

## Author Contributions

**Formal analysis:** Roni Kraut, Christina Korownyk, Danielle Perry, Scott Garrison.

**Investigation:** Roni Kraut, Faith Wierenga, Elisa Molstad, Liz Dennett, Scott Garrison.

**Methodology:** Roni Kraut, Christina Korownyk, Danielle Perry, Scott Garrison.

**Project administration:** Roni Kraut.

**Resources:** Christina Korownyk.

**Writing – original draft:** Roni Kraut, Faith Wierenga.

**Writing – review & editing:** Faith Wierenga, Elisa Molstad, Christina Korownyk, Danielle Perry, Liz Dennett, Scott Garrison.

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
