## [Decision Letter · Decision Letter 0]

27 Sep 2023

PONE-D-23-26403Intolerance upon statin rechallenge: a systematic review and meta-analysis of randomized controlled trialsPLOS ONE

Dear Dr. Kraut,

Thank you for submitting your manuscript to PLOS ONE. After careful consideration, we feel that it has merit but does not fully meet PLOS ONE’s publication criteria as it currently stands. Therefore, we invite you to submit a revised version of the manuscript that addresses the points raised during the review process.

-Authors have selected a very important topic to study. But results need to be explained better. In the results section: we need to report the results with a link to the figure and then talk about them in the discussion.

-Directly in the discussion, authors reported that

*“Only 35*.*9% of participants were intolerant of statins on rechallenge compared to 25.6% who were intolerant on placebo”*

*“Mean myalgia/global symptom score was higher on statins, but only marginally so, and with an upper bound on the 95% confidence interval (3.67 on a 100*-*point scale)”*

Where are these results documented, in which figure. Results need to be reported clearly in the results section with a link to the figure and then we can discuss these results in discussion section.

-Please see reviewer comments below

We look forward to receiving your revised manuscript.

Kind regards,

Vikramaditya Samala Venkata

Academic Editor

PLOS ONE

Journal Requirements:

Reviewers' comments:

Reviewer's Responses to Questions

**Comments to the Author**

1. Is the manuscript technically sound, and do the data support the conclusions?

Reviewer #1: Yes

Reviewer #2: No

Reviewer #3: Yes

2. Has the statistical analysis been performed appropriately and rigorously? 

Reviewer #1: Yes

Reviewer #2: No

Reviewer #3: Yes

3. Have the authors made all data underlying the findings in their manuscript fully available?

Reviewer #1: Yes

Reviewer #2: Yes

Reviewer #3: Yes

4. Is the manuscript presented in an intelligible fashion and written in standard English?

Reviewer #1: Yes

Reviewer #2: Yes

Reviewer #3: Yes

5. Review Comments to the Author

Reviewer #1: -Authors in this article used meta analysis and systematic reviews to know if statins are better tolerated after rechallenge

-Average dose of statin and symptom's upon rechallenge is not clear from the study

Reviewer #2: The objectives of the meta-analysis made good, good review of the available literature including narrowing it down to 8 studies which make inclusion criteria for the meta-analysis. Methods were well explained, and fair explanation of data extraction was performed by the authors. However, the Results of the meta-analysis need to be explained better. The objective of the meta-analysis was to determine tolerance of the statins on reintroduction. The authors do state that only 35.9% of participants were intolerant of statins on rechallenge compared to 25.6% who are intolerant on placebo. It is unclear as to how the authors obtained this result. This needs to be explained in more detail. The data presented in figure 2, 3, 4 does not substantiate this finding. Will appreciate the authors` input.

Reviewer #3: The topic is highly relevant in current practice as the most common question raised by patients while prescribing statin is intolerance. While it is commonly known that most statin intolerance is caused by the nocebo effect, persuading patients to continue treatment can be challenging. Nevertheless, considering the established advantages in preventing cardiovascular incidents, it is crucial to either reintroduce the medication or switch to an alternative form of statin while also considering potential side effects and addressing genuine cases of intolerance.

The study is well done and well written. No correction is recommended.

6. PLOS authors have the option to publish the peer review history of their article (what does this mean?). If published, this will include your full peer review and any attached files.

Reviewer #1: **Yes: **srikanth puli

Reviewer #2: No

Reviewer #3: **Yes: **Nihar Jena

---

## [Author Response · Author response to Decision Letter 0]

8 Nov 2023

Please find below our response to the reviewer feedback in bold. If you require any additional clarification/revisions, please let us know.

1. Authors have selected a very important topic to study. But results need to be explained better. In the results section: we need to report the results with a link to the figure and then talk about them in the discussion.

Directly in the discussion, authors reported that

“Only 35.9% of participants were intolerant of statins on rechallenge compared to 25.6% who were intolerant on placebo”

“Mean myalgia/global symptom score was higher on statins, but only marginally so, and with an upper bound on the 95% confidence interval (3.67 on a 100-point scale)”

Where are these results documented, in which figure. Results need to be reported clearly in the results section with a link to the figure and then we can discuss these results in discussion section.

Thank you for bringing this to our attention. These results are in Figure 2 (meta-analysis of intolerance of statin versus placebo) and Figure 4 (meta-analysis of mean myalgia/global symptom score on statin versus placebo). We have reorganized the results section to make it easier to follow with the figures directly below the discussion. 

2. Average dose of statin and symptom's upon rechallenge is not clear from the study

We decided to provide statin intensity rather than dose in the results section (paragraph 2 and table 1) to make it easier to compare the statin dose between trials. The dose is provided in S2 table: Full data extraction. 

We describe the symptoms measured and the scale used in the results section (paragraph 3 and table 1). The meta-analysis provides the mean and standard deviation of the scores for each study for both placebo and statin (on a 100-point scale).

---

## [Editor Report · Decision Letter 1]

30 Nov 2023

Intolerance upon statin rechallenge: a systematic review and meta-analysis of randomized controlled trials

PONE-D-23-26403R1

Dear Dr. Kraut,

We’re pleased to inform you that your manuscript has been judged scientifically suitable for publication and will be formally accepted for publication once it meets all outstanding technical requirements.

Kind regards,

Vikramaditya Samala Venkata

Academic Editor

PLOS ONE

Additional Editor Comments (optional):

Authors have made the necessary changes. Excellent study and will surely be a great addition to the medical literature.
---

## [Editor Report · Acceptance letter]

11 Dec 2023

PONE-D-23-26403R1 

Intolerance upon statin rechallenge: a systematic review and meta-analysis of randomized controlled trials 

Dear Dr. Kraut:

I'm pleased to inform you that your manuscript has been deemed suitable for publication in PLOS ONE. Congratulations! Your manuscript is now with our production department. 

Kind regards, 

on behalf of

Dr. Vikramaditya Samala Venkata 

Academic Editor

PLOS ONE